# Dose Dependence of Micro-Voids Distributions in Low-Temperature Neutron Irradiated Eurofer97 Steel

**Roberto Coppola [1],\* and Michael Klimenkov [2]**

[1] ENEA-Casaccia, FSN-SICNUC, Via Anguillarese 301, 00123 Roma, Italy
[2] KIT-IAM, Hermann-von-Helmholtz-Platz 1, 76344 Eggenstein-Leopoldshafen, Germany; michael.klimenkov@kit.edu
\* Correspondence: roberto.coppola@enea.it; Tel.: +39-0630484724

**Abstract:** The microstructural effects of mixed spectrum neutron irradiation at 250 °C and 300 °C, for 2.7 dpa, 8.4 dpa, and 16.3 dpa doses, have been investigated in standard Eurofer97 (0.12 C, 9 Cr, 0.48 Mn, 0.2 V, 1.08 W, 0.14 Ta wt%) by means of small-angle neutron scattering (SANS) compared with un-irradiated Eurofer97. The observed SANS effects are attributed to the development of micro-voids, also detected by electron microscopy. The micro-voids distributions have been obtained by an improved transformation method of the SANS cross-sections providing consistent results both before and after subtraction of the un-irradiated reference. Mono-disperse micro-voids distributions are found, with average radii increasing with the dose, namely 4.4 Å for irradiation to 2.7 dpa at 300 °C, 6.6 Å for 8.4 dpa at 300 °C, and 12.9 Å for 16.4 dpa at 250 °C; the corresponding volume fractions are 0.001, 0.006, and 0.004, respectively. The differences in such distributions might reflect different damage evolution mechanisms for the different irradiation conditions, as also suggested by the comparison with a Eurofer97 sample irradiated under fast spectrum. A good resistance of Eurofer97 to micro-structural radiation damage, at least under these irradiation conditions, is suggested by the analysis of these experimental results.

**Keywords:** small-angle neutron scattering; electron microscopy; radiation damage; ferritic/martensitic steels

## 1. Introduction

The low-activation, ferritic/martensitic steel Eurofer97 (0.12 C, 9 Cr, 0.48 Mn, 0.2 V, 1.08 W, 0.14 Ta wt%), European reference for near-term applications in fusion technology, has been extensively characterized by several neutron irradiation experiments [1–5], followed by post-irradiation mechanical testing and by micro-structural examinations, in order to predict its performance in service. This paper refers to the investigation of Eurofer97 samples neutron irradiated in the frame of the SUMO [2,3] and SPICE [4,5] neutron irradiation experiments. The results of post-irradiation mechanical testing and electron microscopy observation of these samples are reported in References [4–8]. In these works, the increase in hardening and embrittlement under neutron irradiation at temperatures lower than 350 °C was tentatively attributed to the observed increase in dislocation loops density. A summary of the information gained by all these irradiation experiments is included in Reference [9], presenting a general assessment of irradiation effects in reduced activation ferritic/martensitic steels, such as Eurofer97: clear trends are observed concerning the impact of neutron irradiation on the mechanical properties but detailed microstructural analysis of neutron irradiated specimens is required for deeper understanding of the nature of the radiation damage in such steels.

Concerning more specifically the microstructural characterization, transmission electron microscopy (TEM) has been utilized to locally investigate the defects produced under the different

irradiation conditions [6–8]. Small-angle neutron scattering (SANS) experiments have also been carried out [10–14], investigating samples issued from the same material utilized for post-irradiation mechanical testing and for TEM. SANS is a useful technique for this task because it samples volumes of approximately 0.1 cm$^3$, yielding micro-structural information more easily comparable with macroscopic properties. Furthermore, due to their magnetic spin, moment neutron beams provide a unique probe of magnetic micro-structural features. Finally, SANS experiments require a very limited sample manipulation, an appreciable feature when having to test highly radioactive samples, like the neutron-irradiated Eurofer97 ones. On the other hand, it is an indirect technique, requiring a mathematical transformation of the data from reciprocal to real space to obtain morphological information on defect size distributions. Therefore, SANS results on complex phenomena, like micro-structural evolution in irradiated steels, must be compared with TEM and all other available metallurgical information.

The investigated samples were made available over several years following the development of the irradiation experiments; therefore they were transported to the neutron source and tested by different SANS experiments. That implied a series of unavoidable experimental uncertainties, even selecting each time the same nominal experimental conditions on the same SANS instrument. Recently, a new series of SANS measurements have been carried out to obtain a more homogeneous data set and quantitatively compare the SANS cross-sections obtained for the different investigated samples [13,14]. In this paper, defect distributions are presented, obtained by improving the previously utilized inverse transformation method of such SANS results. Namely, the fitting parameters have been re-defined taking into account the metallurgical features of the investigated samples and the effect of the background subtraction has been quantitatively characterized, improving the accuracy in the characterization of defects smaller than 10 Å.

This work is therefore intended as a contribution to understand better the microstructural changes in low temperature irradiated Eurofer97 and also because TEM observations of this steel are limited, quantitatively and qualitatively, as mentioned in the next section. Furthermore, an accurate and reliable determination of the defect distribution is necessary also to try and develop theoretical models of radiation damage evolution in such complex materials. As shown in the following two sections, TEM observations and SANS data analysis suggest that the defects originating the observed SANS effects are micro-voids. However, the microstructure of the investigated samples is complex; therefore this is not to be considered as a conclusive interpretation, but as a tentative one, based on the current metallurgical knowledge of the investigated material.

## 2. Material Characterization

The three investigated standard Eurofer97 samples had been neutron irradiated to dose levels of 2.7 dpa (displacement per atom) at 300 °C, 8.4 dpa at 300 °C, and 16.3 dpa at 250 °C. The irradiations had been carried out at the high flux reactor (HFR)—Petten, the first one in the frame of the "SUMO" experiment [2,3] and the two others in the frame of the "SPICE" irradiation experiment [4,5]. Due to their high activity level (20 mSv/h) and to the timing of the irradiation campaigns, the samples had to be transported to the neutron source at different times and consequently included in different SANS experiments. The samples utilized for the SANS measurements were cut from the KLST specimen; after the SANS measurements were completed, they were thinned and prepared for TEM observations. A Eurofer97 sample submitted to standard treatment (980 °C 0.5 h/air + 760 °C 1.5 h/air) was utilized as an un-irradiated reference. All the investigated samples were approximately 1 cm$^2$ in surface area and 1 mm thick; their faces had been electrochemically polished.

TEM observations of these samples [6] showed the presence of dislocation loops and, for the 16.3 dpa dose level, of very few and completely non-homogeneously distributed voids or helium bubbles, with an estimated average size of approximately 70 Å. Such micro-voids could be identified only for irradiation temperatures of at least 350 °C, as shown in Figure 1; probably, for the irradiation temperatures of 250 °C and 300 °C their size and volume fractions are below the resolution capabilities of such observations. Furthermore, the TEM observation of these voids, embedded in a magnetic

matrix, was very difficult due to defocusing effects and to their small size; even at 350 °C the detected number of them was not sufficient to provide a size histogram.

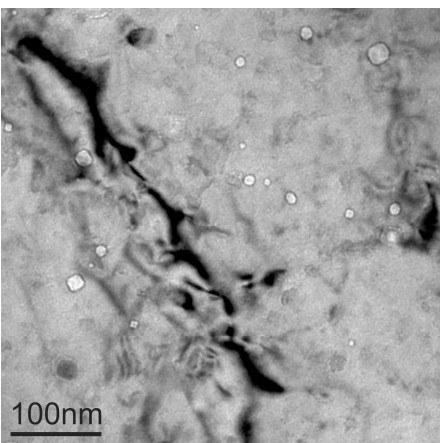

**Figure 1.** Bright field TEM observation of micro-voids in Eurofer97 steel neutron irradiated at 350 °C to 16 dpa.

## 3. Experimental Technique

General information on SANS can be found in References [15,16]. For studying ferritic/martensitic steels, an external magnetic field of at least 1 T is needed to saturate the magnetization in the sample and separate the nuclear and the magnetic SANS cross-sections, as shown in Figure 2. The total SANS cross-section is defined as follows:

$$d\sum (Q)/d\Omega = d\sum (Q)/d\Omega_{\text{nucl}} + d\sum (Q)/d\Omega_{\text{mag}} \sin^2 \alpha \tag{1}$$

where $2\theta$ is the full scattering angle, $\lambda$ the neutron wavelength, $Q = 4\pi\sin\theta/\lambda$ the modulus of the scattering vector, $\alpha$ the azimutal angle on the detector plane and $\Omega$ stands for the solid angle. The ratio of the SANS components measured in the directions perpendicular and parallel to the magnetic field, defined as follows:

$$R(Q) = \frac{d\sum (Q)/d\Omega_{\text{nucl}} + d\sum (Q)/d\Omega_{\text{mag}}}{d\sum (Q)/d\Omega_{\text{nucl}}} = 1 + (\Delta\rho)^2_{\text{mag}}/(\Delta\rho)^2_{\text{nucl}} \tag{2}$$

is related to the nuclear and magnetic square differences in neutron scattering length density between the defects and the matrix, $(\Delta\rho)^2_{\text{nucl}}$ and $(\Delta\rho)^2_{\text{mag}}$, respectively [15,16].

It can provide information useful to identify the scattering defects: namely, for non-magnetic defects embedded in a fully magnetized matrix, it is equal to 2, while its dependence on $Q$ can imply the presence of different kinds of defects. In Eurofer97, assuming for the carbides a composition $Cr_{14}Fe_8W_{0.7}V_{0.3}C_6$ [17] a nuclear contrast value of $2.13 \times 10^{20}$ cm$^{-4}$ is found for such precipitates; for micro-voids, the contrast is equal to the scattering length density of Eurofer97-1 itself, that is $5.51 \times 10^{21}$ cm$^{-4}$, while for helium bubbles it is $4.88 \times 10^{21}$ cm$^{-4}$ [16]. Therefore, for comparable values of the corresponding volume fractions, both helium bubbles and micro-voids are expected to give rise to SANS effects one order of magnitude larger than precipitates, however, they are quite difficult to be distinguished from one another. A summary of the SANS set-up and experimental conditions selected to obtain the results analyzed in this paper is included in the next section.

The distributions of the scattering defects are obtained by inverse transformation of the experimental data. Namely, if their volume fraction is low and there is no inter-particle interference, the SANS nuclear and magnetic cross-sections can each be written as:

$$d\sum(Q)/d\Omega = (\Delta\rho)^2 \int_0^\infty dR N(R) V^2(R) |F(Q,R)|^2 \tag{3}$$

where $N(R)$ is the number per unit volume of defects with a size between $R$ and $R + dR$, $V$ their volume, $|F(Q,R)|^2$ their form factor and $(\Delta\rho)^2$ is the nuclear or magnetic "contrast".

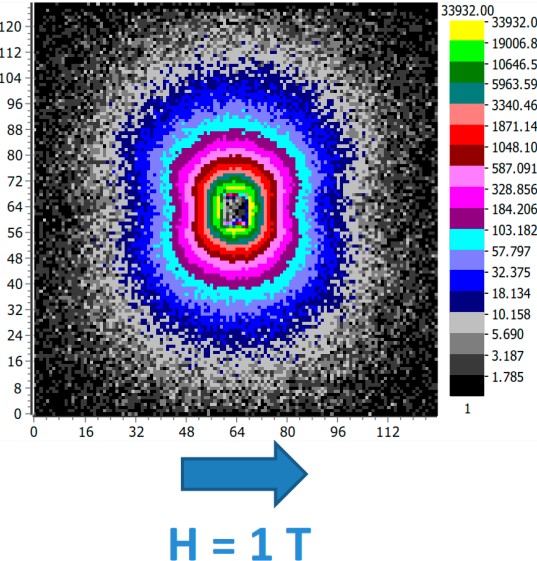

**Figure 2.** 2D small-angle neutron scattering (SANS) pattern (neutron counts) of the un-irradiated Eurofer97 reference sample, measured with $\lambda = 6$ Å at 2 m sample-to-detector distance; external magnetic field direction is horizontal in the plane of the figure.

The volume distribution function $D(R)$, average defect radius, $<R>$, and volume fraction, $f$, are defined respectively as follows:

$$D(R) = N(R)R^3 \tag{4}$$

$$<R> = \int_0^\infty dR N(R) R / \int_0^\infty dR N(R) \tag{5}$$

$$f = \int_0^\infty dR N(R) V(R) / (\Delta\rho)^2 V_{\text{tot}} \tag{6}$$

where $V_{\text{tot}}$ is the total volume of the investigated sample.

$N(R)$ was determined by the method described in Reference [18] and successfully utilized for several studies on technical steels, particularly those in References [19–21]. This fitting procedure assumes no a priori shape of the defect distribution, representing it by a set of cubic B-spline functions, with knots uniformly distributed in a log $R$ scale and with the constraint $N(R) > 0$. The number of splines is chosen taking into account the $R$-range $R_{\text{min}} - R_{\text{max}}$ where the size distribution has to be investigated and the shape of the experimentally determined SANS cross-section. Knots spacing, $R$-range, and a constant or $Q$-dependent background are additional parameters, adjustable for improving the fit. The best-fit distribution is determined within an 80% confidence band. In general, the main difficulties in obtaining the distributions from Equation (3) arise from the fact that the SANS cross-section is measured only on a finite $Q$-interval, in a limited number of points, affected by experimental errors. Furthermore, a theoretical model distribution usually is not available for irradiation effects in such technical steels. All this must be taken into account in selecting the fitting parameters, to obtain plausible distributions and, at the same time, reduce their error band as much as possible.

## 4. Results and Discussion

The SANS measurements were carried out utilizing the D22 instrument at the Institut Max von Laue-Paul Langevin in Grenoble [22]. First the samples irradiated to 2.7 dpa at 300 °C and to 8.4 dpa at 300 °C were investigated [10,11]; subsequently, the sample irradiated to 16.3 dpa at 250 °C was investigated [12]. To clarify some uncertainties relating to the utilized un-irradiated reference samples [12] and to obtain a more homogeneous data set in the same experimental conditions, the two samples irradiated at the higher dose levels were recently measured again, together with other similar ones [13,14]. A neutron wavelength of 6 Å and sample-to-detector distances of 2 m and 11 m were selected in order to cover a $Q$-interval ranging from $3.0 \times 10^{-3}$ Å$^{-1}$ to $2.6 \times 10^{-1}$ Å$^{-1}$, corresponding to sizes $2R \sim \pi/Q$ varying between 10 Å and 1000 Å approximately. An external magnetic field of 1 T was applied in the horizontal direction, perpendicular to the neutron beam (see Figure 2). The SANS cross-sections parallel and perpendicular to the applied magnetic field were determined by selecting on the 2D detector plane sectors 15° wide. Each experiment included the measurement of the same un-irradiated Eurofer97 reference sample, to check that the cross-sections of the irradiated samples, obtained in the different experiments, can be quantitatively compared after calibration into physical units. Figure 3 shows the nuclear SANS cross-sections of the reference sample as measured in the first experiment [10] and in the latest one [13]: they are in good agreement, also considering that SANS data had been reduced by two different programs, namely the one described in Reference [23] for the first experiment and the one described in Reference [24] for the latest experiment. The experimental errors on the SANS cross-sections were of a few % in all these experiments.

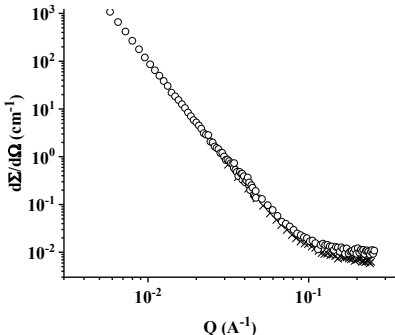

**Figure 3.** Nuclear SANS cross-sections for Eurofer97 un-irradiated reference sample from References [10] (crosses) and [14] (empty circles).

These experimental results are summarized in the overview of Figure 4.

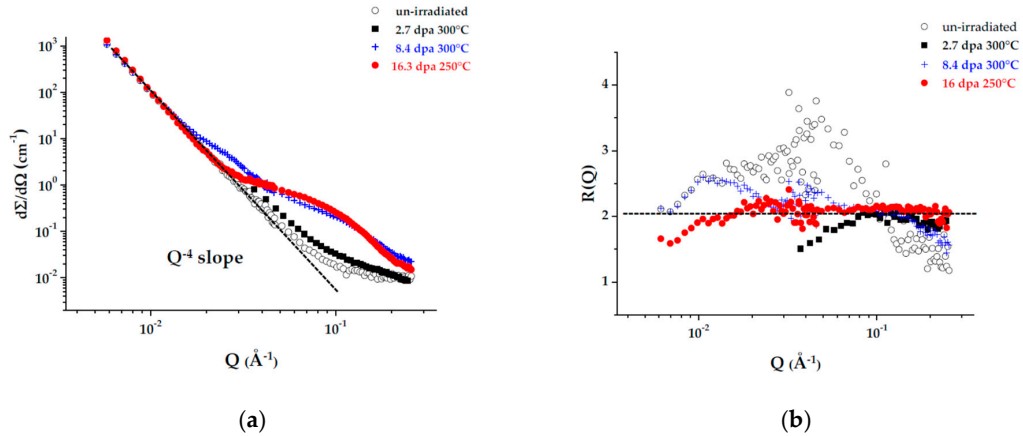

|(**a**)|(**b**)|

**Figure 4.** Nuclear SANS cross sections (**a**) and *R(Q)* (**b**) for un-irradiated reference Eurofer97 (empty circles) and for Eurofer97 irradiated 2.7 dpa 300 °C (squares), 8.4 dpa 300 °C (crosses), 16.3 dpa 250 °C (full circles); experimental results from References [10,14].

The nuclear SANS cross-sections of irradiated and un-irradiated reference samples are shown separately for each irradiation condition in Figure 5 to appreciate the changes with respect to the reference sample more adequately and also because of the differences in $Q$ range and cross-sections between the 2.5 dpa irradiated sample and the two others. Similar differences are observed when comparing the nuclear plus magnetic SANS cross-sections of these same samples.

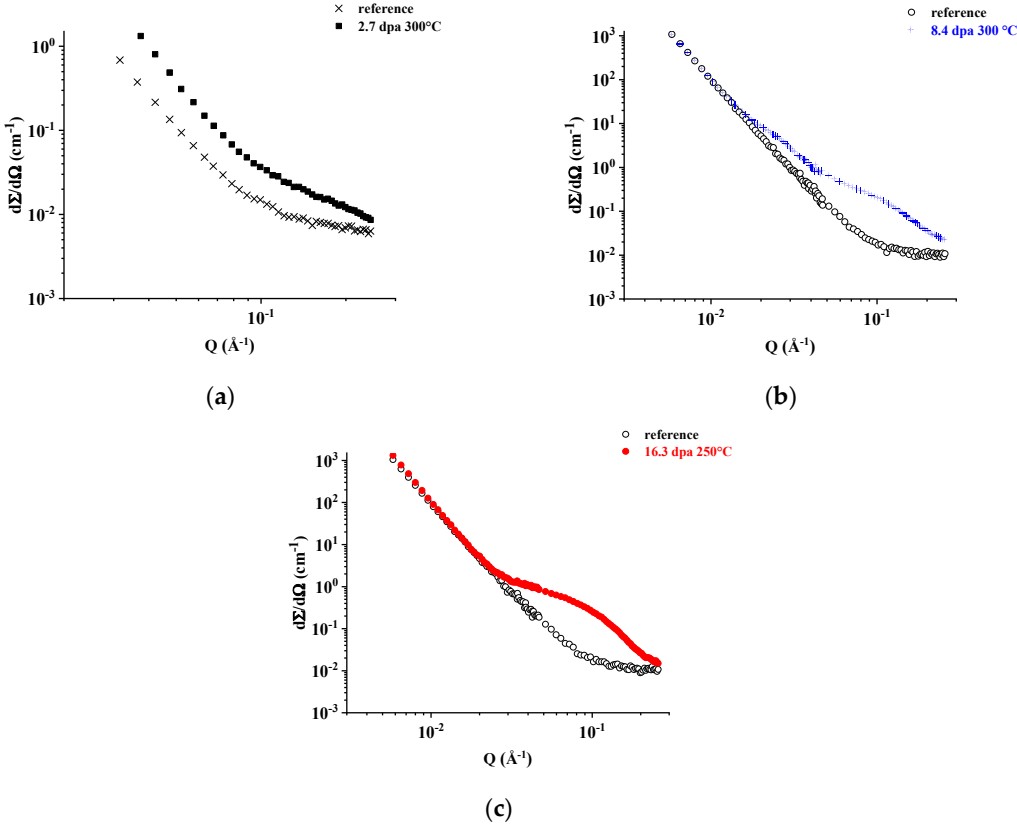

**Figure 5.** Nuclear SANS cross-sections for Eurofer97: (**a**) irradiated 2.7 dpa 300 °C (squares) and reference (*x* crosses) [10], (**b**) irradiated 8.4 dpa 300 °C (+ crosses) and reference (empty circles) [14], (**c**) irradiated 16.3 dpa 250 °C (full circles) and reference (empty circles) [14].

A small but well detectable increase in the SANS cross-section is observed after irradiation to 2.5 dpa. In the same $Q$ range, for the samples irradiated to 8.4 dpa and 16.3 dpa the increase in SANS cross-section is one order of magnitude higher. Furthermore, the wider $Q$ range available for the measurements of these two samples shows that for approximately $Q < 10^{-2}$ Å$^{-1}$ their cross-sections are nearly coincident with one of the reference samples and all follow a $Q^{-4}$ behavior ("Porod law" [15,16], see also Figure 4a), determined by defects with sizes outside the experimental $Q$ window: this behavior is attributed to carbide precipitates larger than approximately 500 Å, generally not affected by irradiation for such temperatures and dose levels [25]. As shown in Figure 4b, for the three irradiated samples the $R(Q)$ ratio is close to 2 and within the experimental uncertainties introduced by combining the nuclear and nuclear plus magnetic SANS cross-sections as indicated in Equation (2). The slightly higher $R(Q)$ value measured for the sample irradiated to 8.4 dpa around approximately $Q = 2.5 \times 10^{-2}$ Å$^{-1}$, in correspondence to the small "bump" visible in Figures 4a and 5b, could be attributed to the presence of a different kind of defect.

The size and volume distribution functions have been determined assuming that the observed SANS effects are determined by micro-voids as those shown in Figure 1; therefore, taking a nuclear contrast value of $5.51 \times 10^{21}$ cm$^{-4}$, a spherical form factor was assumed. In the previous papers [10–14], such distributions were obtained introducing up to 10 spline functions in order to fit the whole SANS cross-sections at least in the higher $Q$-range and corresponding to 2 m sample-to-detector distance.

Furthermore, a wide $R_{min} - R_{max}$ range had been selected with $R_{min} = 1$ Å. This approach yielded defect distributions affected by unphysical oscillations and by a large error band, with consequent difficulties in attempting a metallurgical interpretation: this is particularly evident in Figure 2 of Reference [14]. On the other hand, recent work carried out on SANS data obtained from other irradiated Eurofer97 samples [21] has shown that more accurate and significant distributions are obtained reducing the number of splines to a maximum of 7, setting $R_{min}$ to the more realistic value of 3 Å and excluding from the fitting procedure those points clearly affected by the "Porod background" at low $Q$'s and by uncertainty on the background level at high $Q$'s. A constant background of few units in $10^{-3}$ cm$^{-1}$ has also been introduced to improve the fit for the points at the higher $Q$ values, particularly for the data in Figure 5c. This way to proceed implies an unavoidable degree of arbitrariness, either in the choice of the fitting parameters or in the selection of the points to be excluded from the fit; however, no theoretical model is currently available for such complex samples, therefore at present only an empirical SANS data analysis seems possible. The data shown in Figure 5 have been analyzed in this way, determining for each irradiated sample the best-fit distributions obtained with and without the subtraction of the reference sample's cross-section. As the SANS cross-section (Equation (1)) is governed by the square volume of the defects, volume distributions provide metallurgical information that is more significant with respect to the number of distributions, as shown in previous research [21]. Therefore, the obtained volume distributions as defined in Equation (5) are presented here below.

For the sample irradiated at 300 °C to 2.7 dpa, a good fit with the experimental SANS cross-section is obtained both with and without the subtraction of the reference sample's cross-section (Figure 6a); in both cases, it is necessary to exclude from the fit a few points at low $Q$'s, probably partly affected by the "Porod background." The corresponding distributions are shown in Figure 6b: subtracting the reference sample yields a mono-disperse distribution of very small voids (*<R>* = 4.4 Å) with a volume fraction of 0.001. Considering that interatomic distances in such steels are approximately 3 Å, micro-voids of approximately 10 Å in diameter should be considered as very small vacancy clusters that are close to the experimental resolution limit of the SANS and TEM techniques.

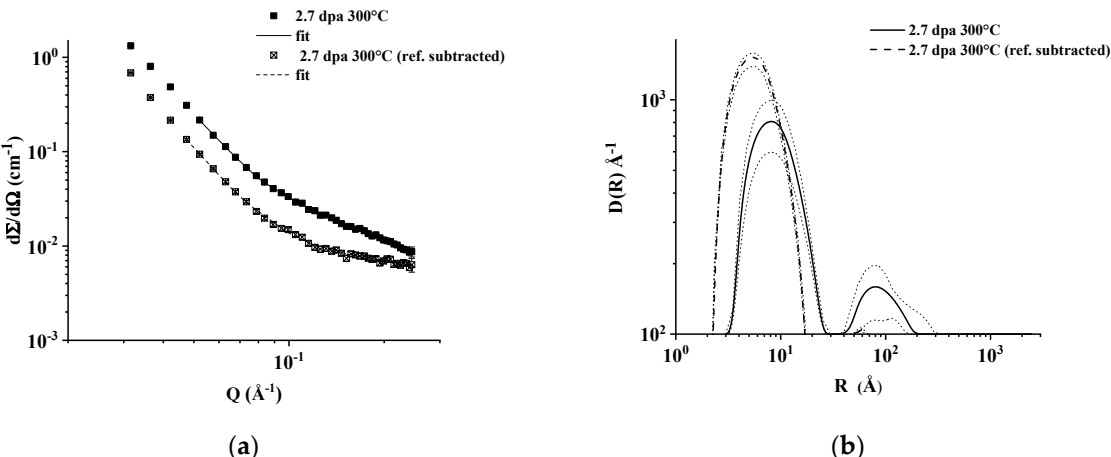

**Figure 6.** Eurofer97 neutron irradiated to 2.7 dpa at 300 °C: (**a**) best fits to nuclear SANS cross-section without (full squares) and with (empty squares) subtraction of the reference sample's cross-section, (**b**) best-fit micro-void distributions (Å$^{-1}$) obtained without (continuous line) and with (dashed line) subtraction of reference sample's cross-section. The 80% confidence bands are indicated by the dotted lines.

Also for the sample irradiated at 300 °C to 8.4 dpa a good fit is obtained both with and without subtraction of the reference sample's cross-section (Figure 7a). The corresponding distributions (Figure 7b) are very similar, with volume fraction and average radius values around 0.007 and 6.5 Å respectively. A secondary population of defects one order of magnitude larger is also observed after subtraction of the reference sample: it is correlated with the cross-section's trend in the $Q$ range

approximately $3.5$–$2 \times 10^{-2}$ Å$^{-1}$ and its size corresponds to the larger micro-voids shown in Figure 1 and in Reference [6]. However, as pointed out above, the $R(Q)$ value is significantly higher than 2, suggesting the presence of a different kind of defect.

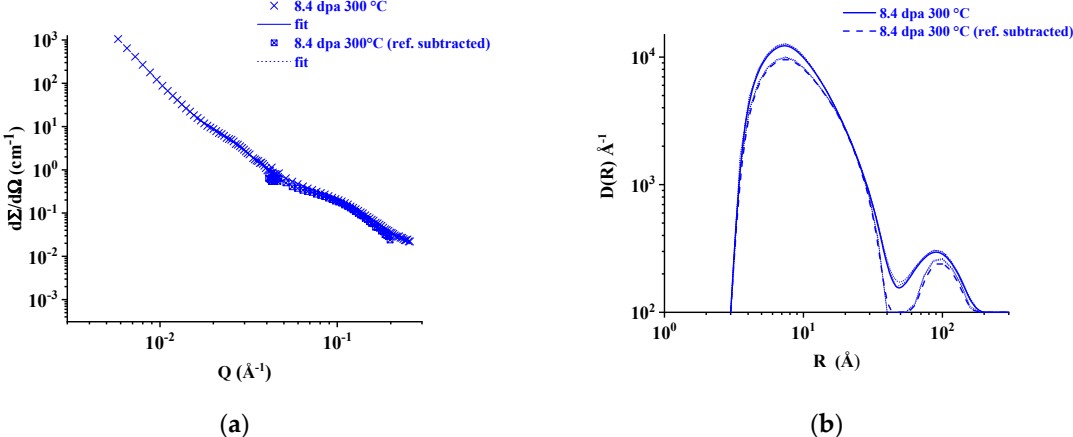

(a)　　　　　　　　　　　　　　　　　　　　　　　　　　　(b)

**Figure 7.** Eurofer97 neutron irradiated to 8.4 dpa at 300 °C: (**a**) best fits to nuclear SANS cross-section without (crosses) and with (empty squares) subtraction of reference sample's cross-section, (**b**) best-fit micro-void distributions (Å$^{-1}$) obtained without (continuous line) and with (dashed line) subtraction of reference sample's cross-section. The 80% confidence bands are indicated by the dotted lines.

For the sample irradiated at 250 °C to 16.3 dpa, the subtraction of the reference's sample cross-section is possible only in a restricted $Q$ range, where the effect of the irradiation is detected (Figure 5c), but a good fit is anyhow obtained (Figure 8a). Similar mono-disperse distributions are obtained (Figure 8b) with volume fraction and average radius values around 0.004 and 12.9 Å, respectively.

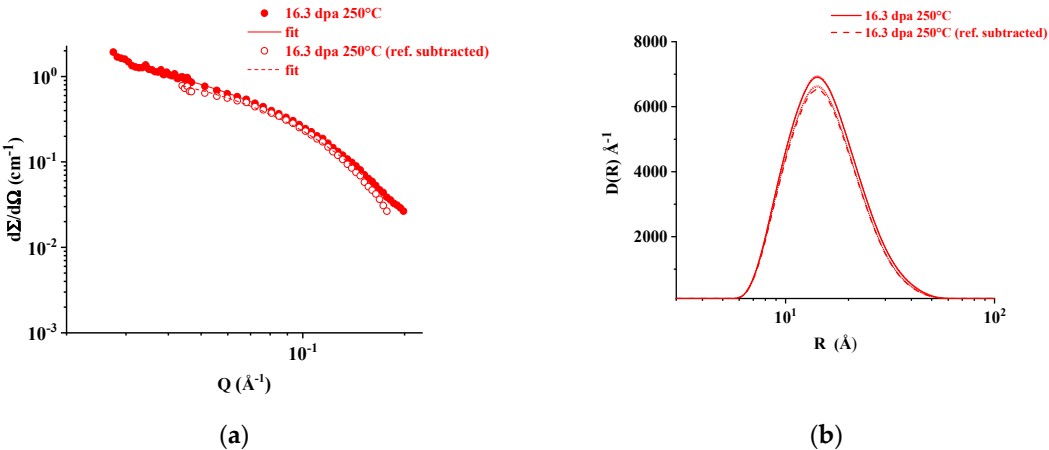

(a)　　　　　　　　　　　　　　　　　　　　　　　　　　　(b)

**Figure 8.** Eurofer97 neutron irradiated to 2.7 dpa at 300 °C: (**a**) best fits to nuclear SANS cross-section without (full circles) and with (empty circles) subtraction of reference sample's cross-section, (**b**) best-fit micro-void distributions (Å$^{-1}$) obtained without (continuous line) and with (dashed line) subtraction of reference sample's cross-section. The 80% confidence bands are indicated by the dotted lines.

The distributions obtained for these three samples after reference subtraction are compared in Figure 9; Table 1 lists the obtained average radii and volume fractions. The uncertainty bands visible in Figures 6–8 and the corresponding errors on the values of Table 1 are smaller than 10%. These errors result from the best fits obtained under the assumptions described here above. An estimate of the uncertainties resulting from different methodological assumptions and choice of the fitting parameters is provided for example by the comparison with the distributions previously obtained for these same samples [14]. Consistently higher values of volume fraction and lower values of average radii had

been obtained with the already mentioned larger error bands and unphysical oscillations (Figure 2 in Reference [14]): in fact in the previous works, the cross-section considered for the fit was not solely determined by the effect of the irradiation, but partly contaminated by the spurious effects described here above. Furthermore, the previous SANS data analysis lead to some incorrect conclusions about the shape of the distributions, appearing bi-modal as particularly evident in Figure 4 of Reference [11], with no change in the average radii; however, this result was obtained because spurious effects related to the background noise and asymptotic contribution of large carbide precipitates were included in the fit. On the contrary, genuine information on the irradiation effects is provided by the new distributions presented in this paper, summarized in Figure 9. They are mono-disperse and the micro-void radius increases with the dose. In fact, after subtraction of the reference samples, an average radius of 4.4 Å is found for 2.7 dpa, 6.6 Å for 8.4 dpa, and 12.9 Å for 16.3 dpa. Furthermore, the previous SANS data analysis did not allow any clear conclusion on the effect of subtracting the reference sample's cross-section nor a reliable determination of the average radii, while the volume fractions were overestimated. These problems are solved by the new data analysis presented in this paper and summarized in Table 1.

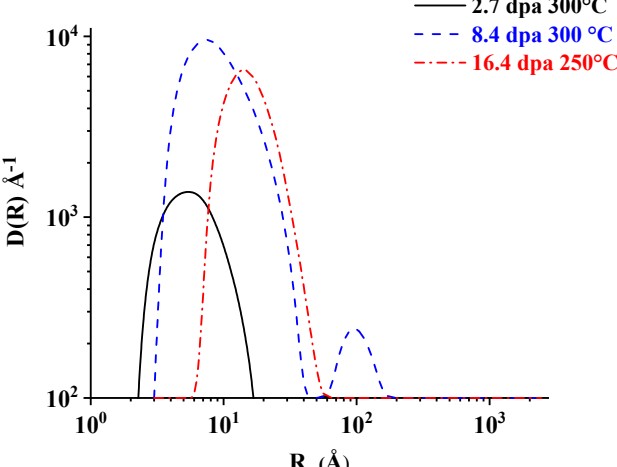

**Figure 9.** Best-fit micro-voids volume distributions for Eurofer97 neutron irradiated to 2.7 dpa at 300 °C (continuous line), 8.4 dpa at 300 °C (dashed line), 16.3 dpa at 250 °C (dash-dotted line). The distributions are those shown in Figures 6b, 7b and 8b, respectively after reference sample subtraction.

**Table 1.** Micro-voids volume fractions and average radii in neutron irradiated Eurofer97; the values obtained after subtracting the un-irradiated reference sample's cross-section are marked in parenthesis and italics.

| Sample | Volume Fraction, $f$ | Average Radius, $<R>$ |
|---|---|---|
| 2.7 dpa 300 °C | 0.002 (0.001) | 6.8 Å (4.4 Å) |
| 8.4 dpa 300 °C | 0.007 (0.006) | 6.5 Å (6.6 Å) sec. ~100 Å |
| 16.4 dpa 250 °C | 0.005 (0.004) | 12.8 Å (12.9 Å) |

The obtained results suggest first of all that up to 16.3 dpa only low volume fractions of small micro-voids are produced in Eurofer97 under neutron irradiation at low temperature; this is consistent with resistance to irradiation expected for this steel [9]. It is difficult to establish clear trends of microstructural damage evolution with dose or temperature, also because only three irradiated samples were investigated—although even for such a limited number a consistent effort was needed for sample transportation and experiment organization, given the high activity of the samples. However, a significant increase of the micro-voids average radius with the dose is observed and, most importantly, well defined mono-disperse distributions are found after subtraction of the reference sample. This is potentially quite

useful for modeling purposes of radiation damage evolution in this steel. The secondary population observed in the sample irradiated to 8.4 dpa, consistent with the measured SANS cross-section could be originated by a different kind of defect (see also Figure 4b) and in this case, the volume distribution should be determined again introducing a model with two different contrast values for two different kinds of defects. No such evidence is provided by the currently available TEM observations [6].

Additional information on the microstructural evolution in low-temperature neutron irradiated Eurofer97 is obtained comparing the distributions of these samples with the one obtained for a Eurofer97 sample irradiated to 32 dpa at 330 °C at the BOR60 reactor [13]; also, in this case, the observed SANS effects are tentatively attributed to micro-voids. Figure 10 shows the volume distribution of this sample re-calculated by the procedure described here above (without reference subtraction) and compared to the distributions shown in Figure 9 for HFR irradiation at 8.4 dpa and 16.4.

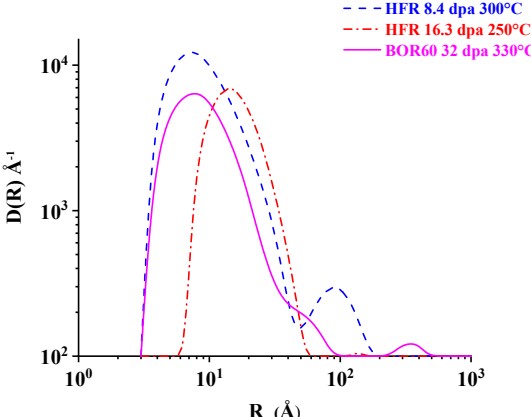

**Figure 10.** Best-fit micro-voids volume distributions of Figure 9 for Eurofer97 neutron irradiated to 8.4 dpa 300 °C (dashed line) and 16.3 dpa 250 °C (dash-dotted line) compared with best-fit distribution of Eurofer97 neutron irradiated at BOR60 reactor to 32 dpa at 330 °C (corresponding SANS measurements shown in Reference [13]).

Despite the higher irradiation dose, the obtained average radius (6.5 Å) and volume fraction (0.007) of the BOR60 sample are the same as those found for the 8.4 dpa HFR irradiated sample. It is noted that also in this case, the previous SANS data analysis yielded unclear conclusions concerning the comparison between the HFR and the BOR60 irradiated samples (Figure 6 Reference [13]). In fact, a more appropriate choice of the fitting parameters provides a well-defined mono-disperse distribution also for the first one. As suggested in Reference [13], the very low helium production under fast neutron irradiation could partly explain this effect because the helium produced under mixed spectrum plays an important role in stabilizing the micro-voids and promoting their growth. A more systematic investigation of samples irradiated for the same dose and temperature at the two neutron sources would be necessary to come to some conclusions. It is noted that the BOR60 sample also shows a secondary population of very large defects (in addition to a mono-disperse distribution) suggesting that this effect related to the higher irradiation temperature for the higher dose levels; in fact, it was not observed in the sample irradiated to 2.7 dpa at 300 °C, although it was measured in a more limited $Q$ range.

## 5. Conclusions

SANS measurements carried out on three different neutron irradiated Eurofer97 steel have been analyzed by improving a previously utilized inverse transformation method. The spurious effects determined by large carbides and by background noise have been evaluated both by subtracting the reference sample's cross-section and by excluding from the fitting procedure the experimental points contaminated by such effects. Consistent results have been found in these two ways; information as genuine as possible has been obtained on the microstructural effect of the irradiation, with more

accurate and realistic distributions potentially useful also for theoretical modeling. Furthermore, these methodological improvements of the SANS data transformation procedure allow for a clearer comparison of the micro-void distributions produced under the mixed and fast spectrum.

Mostly based on the TEM results, the observed SANS effects and obtained distributions are attributed to low volume fractions of small micro-voids, probably below the resolution capability of TEM in a magnetic matrix. An increase in micro-void size with irradiation dose is observed with a well-defined mono-disperse distribution (Figure 9). However, the higher micro-void volume fraction is found for the sample irradiated at the intermediate dose of 8.4 dpa, suggesting the presence of a secondary population of defects as large as the micro-voids shown in Figure 1; the information currently provided by TEM does not allow any conclusion on this specific point. The comparison with another Eurofer97 sample irradiated at 330 °C (Figure 10) suggests that this effect might be related to the higher irradiation temperature. These experimental results also show that the influence of the neutron irradiation spectrum must be thoroughly investigated to understand radiation damage evolution in this steel fully. Work is underway to characterize, by new SANS measurements, Eurofer97 irradiated at 250 °C for 16 dpa at HFR and at BOR60 neutron sources.

The detected SANS effects, shown in Figure 4; Figure 5, are quite clear and the obtained distributions are consistent with them. With respect to the previous efforts to analyze such SANS effects, significant improvements have been obtained particularly concerning the shape of the distributions and their metallurgical interpretations; the results presented in this manuscript provide a new insight into microstructural radiation damage evolution in low temperature irradiated Eurofer97 steel. The microstructure of the investigated samples is very complex; therefore the proposed interpretation in terms of micro-void development is intended as a tentative one based on the metallurgical information currently available. Concerning the correlation between post-irradiation mechanical testing [4–9] and microstructural observations, it would be quite unsafe to make general conclusions based on the SANS analysis of only three irradiated samples of such a complex steel. In any case, taking into account that macroscopic sample volumes have been investigated, these SANS measurements confirm the good resistance of Eurofer97 to irradiation for the considered dose and temperature range.

**Author Contributions:** Conceptualization, R.C. and M.K.; methodology, R.C. software, R.C.; validation, R.C. and M.K.; formal analysis, R.C.; investigation R.C. and M.K.; resources, R.C.; data curation, R.C.; Writing—Original Draft preparation, R.C. Writing—Review and Editing, R.C.; visualization, R.C. and M.K.; supervision, R.C.; project administration, R.C.; funding acquisition, R.C.

**Funding:** EUROfusion Consortium grant agreement No. 633053.

**Acknowledgments:** This work has been carried out within the frame of the EUROfusion Consortium and has received funding from the Euratom research and training programme 2014–2018 and 2019–2020 under grant agreement No. 633053. The views and opinions expressed herein do not necessarily reflect those of the European Commission. R. Lindau (KIT) is gratefully acknowledged for preparing the irradiated sample and organizing their transportation back and forth to the neutron source. M. Valli (ENEA) is acknowledged for informatics support.

**Conflicts of Interest:** The authors declare no conflict of interest.

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
