# Peer review of "Dose Dependence of Micro-Voids Distributions in Low-Temperature Neutron Irradiated Eurofer97 Steel"

_metals, doi:10.3390/met9050552_

Round 1
Reviewer 1 Report
In the composition of the EUROFER 97 steel, the content of Mn and Ta are missing. Can the authors add some more information on the material that is being studied?
Figure 1 is confusing: it shows a TEM picture of a sample that was not investigated with SANS. If it's used as a reference, it's better to put it in the discussion when explaining the SANS results. Or better: can’t it be replaced with TEM pictures of the 3 samples that were investigated here? Especially the sample irradiated to 8.4 dpa would be interesting to show. Other neutron irradiated EUROFER97 samples (eg. Weiss, Gaganidze, Aktaa, J.Nucl. Mater. 426 (2012) 52) do not have such large cavities. It will strengthen the paper if the authors can show large cavities in the sample irradiated to 8.4dpa and not in the other two samples.
I disagree with the statement in line 84 that is was not possible to provide a statistically significant size histogram, it requires more work to get it. However, the size distribution of the sample in figure 1 would be irrelevant as it was not studied with SANS here.
Line 170: There is no figure 1a in this paper, should that be figure 4a?
Line 173: the statement of the R(Q) ratio being different from 2 at Q=2.5×10^-2 A-1 does not agree with figure 4b in which higher R(Q) ratios can be found at lower and higher Q-values, but not in the area of the “bump”. Moreover, if the defects causing the bump are identified as larger voids, they are non-magnetic and should have a ratio of 2.
Ref 13: the authors of that paper are not correctly represented
Reviewer 2 Report
The manuscript reports on the dose dependence of micro-voids distributions in low temperature neutron irradiated Eurofer97 steel.
From the presentation perspective, the manuscript is clearly written and the results an be promptly followed. The authors state in the introduction that they have applied the technique to analyze the said material with some long timeframe (completely fine since beamtime takes time to obtain).
However, the manuscript does not state clearly what is the novelty that it brings to its audience, since the authors already published a substantial amount of data in the topic. As stated in the conclusion "... by improving a previously utilized inverse transformation method". This sentence indicates to the reader that the reported activity is a continuation of previous efforts, with no clear new insight/methodology.
In view of this perspective, I do not recommend it for publication as it is.
Reviewer 3 Report
This paper looks to characterise void formation due to irradiation exposure by using SANS as the primary method. It draws upon previous work and re-analyses the SANS with consideration of information drawn from other analysis techniques (TEM).
The conclusion of this article is that the volume fraction of the voids increases 0.1% and by 0.1A besides the lowest dose which increases by 2.4A yet the volume fraction still only increases by 0.1%. This is then attributed to unknown defects.
This paper aims to improve the analysis of already existing SANS spectra. The benefits and accuracy of this new method are not made clear. For this appear to be published it needs to concentrate on the improvement on the analysis in comparison to existing analysis rather than presenting the findings as if a new result.
The first paragraph needs to discuss what in in the references mentioned, and relate this to the work in this paper. The introduction needs to more clearly state what this research is adding, why it is important, and how it relates to existing research in a clearer manner.
Experimental technique needs to give a brief summary of the SANS setup rather than just refer to other papers i.e. move the description from line 169-174 into the experimental procedure section.
Figure 4a and Figure 5 show the same data. Figure 3 and 4 are not discussed in the paper. Many of the results figures are not discussed in the paper.
A second unknown defect is discussed but not pursued yet conclusions are drawn from it.
Round 2
Reviewer 3 Report
The clarity of statements made in the response to reviewer should be incorporated into the paper. The responses to reviewer concisely identify and then address the issues raised. If these type of statements were included in the manuscript it would strengthen the article's story and argument.
The concise statements provided in the response to reviewer also when collected would improve the conclusion of the article.
If these sentiments are more clearly stated in the article I consider it acceptable for publication.
Author Response
Please, see attached document.
